# Effect of Mechanical Shock Treatment on Microstructure and Corrosion Properties of Manual Argon Arc Welding Joints of 2205 Duplex Stainless Steel

**DOI:** 10.3390/ma15093230

**Published:** 2022-04-29

**Authors:** Lingqing Gao, Xinyao Zhang, Xiaoqin Zha, Xinyu Zhang

**Affiliations:** 1Luoyang Ship Material Research Institute, Luoyang 471023, China; gaolq@163.com (L.G.); xiaoqinzha@sina.com (X.Z.); 18629660516@163.com (X.Z.); 2Henan Key Laboratory of Technology and Application of Structural Materials for Ships and Marine Equipments, Luoyang 471023, China

**Keywords:** 2205 duplex stainless steel welded joint, mechanical shock treatment, microstructure, corrosion properties

## Abstract

Pneumatic chipping hammer and ultrasonic impact peening were used to relieve the welding residual stress of 2205 duplex stainless steel by manual argon arc welding, and the influences of these mechanical shock treatment technologies on the residual stress, microstructure, and corro-sion resistance of the welding seam were studied. Results showed that after pneumatic chipping hammer or ultrasonic impact peening, a small amount of plastic deformation occurred in the welded joint of 2205 duplex stainless steel, which led to an increase in the dislocation density in the microstructure. Meanwhile, the stress state of the welded joint changed from the residual tensile stress to the residual compressive stress. The maximum residual compressive stress could reach −579 MPa. The combined action of the two effectively improved the corrosion resistance of the welded joint. Among them, the best overall effect was the ultrasonic impact peening tech-nology.

## 1. Introduction

As is known, the yield strength of 2205 duplex stainless steel is twice that of austenitic stainless steel due to the combination of the excellent properties of ferrite and austenite [1,2,3]. Meanwhile, it has the strong chloride corrosion resistance and stress corrosion resistance of ferritic stainless steel [4,5,6]. Excellent service performance makes it widely used in energy, chemical industry, construction, oil and gas transportation, and other fields. Welding is very important in the industrial production process of duplex stainless steel pipes, plates, and storage containers [7,8]. Since the microstructure and mechanical properties of the welded joint are quite different from the base material, the quality of the welded joint directly determines the service life of the 2205 duplex stainless steel workpiece.

At present, the welding process of 2205 duplex stainless steel has been studied widely. Badji et al. [9] studied the phase transformation and mechanical behavior of 2205 duplex stainless steel during welding and subsequent annealing. The results showed that the content of ferrite was higher in the heat affected zone, while the content of austenite was higher in the weld zone. In the annealing temperature range of 800–1000 °C, σ phase and M_23_C_6_ chromium carbides are precipitated at the γ/δ interface. When the annealing temperature exceeds 1050 °C, the volume fraction of δ ferrite rises with the increase in the annealing temperature. Under the annealing condition of 1050 °C, the mechanical properties of the whole weld zone are the best. Daniel et al. [10] studied the laser weld of SAF 2205 duplex stainless steel plate at different welding speeds, and tested the microstructure and mechanical properties of the welded joints. The results showed that at different welding speeds, the content of ferrite in the weld zone was higher than 85%. In the base material, there was a significant elemental partition between ferrite and austenite. However, no element distribution was found in the weld zone due to the very high cooling rate limiting the possible diffusion. The maximum hardness of the fusion zone was 310 HV_0.2_, which was 17% higher than the base material hardness, and rose with the increase in welding speed. However, the welding speed had no significant effect on the yield strength and tensile strength. Kordatos et al. [11] studied the effect of cooling rate on the mechanical properties and corrosion properties of the 2205 duplex stainless steel weld. The results show that the faster cooling rate reduces the width of the ferritized zone of the duplex stainless steel weld from 600 μm in air cooling to 500 μm in water cooling. In the water-cooled welds, the austenite volume fraction in the weld zone decreased slightly, but could still reach 43%, which provided good corrosion resistance and mechanical properties for the welds. The hardness of the weld zone was affected by the cooling rate. Because the water-cooled sample had a higher ferrite volume fraction than that of the air-cooled sample, its hardness was higher and the toughness was also reduced. These existing research results provide certain technical support for the development of the duplex stainless steel welding process.

However, during the welding process, the weld area of the duplex stainless steel will form a large residual tensile stress field, which not only causes the deformation of the welded joint, but also has a negative impact on the corrosion resistance and fatigue life of the welded component [12,13,14]. Therefore, it is of great significance to optimize the performance of welded joints to improve the residual stress state through mechanical shock treatment technology. At present, there are few reports on the effect of mechanical shock treatment on the microstructure and properties of 2205 duplex stainless steel welded joints. In the present study, the microstructure analysis and corrosion performance test of the 2205 duplex stainless steel welded joint [15,16,17] by manual argon arc welding after two mechanical shock treatment technologies (Pneumatic chipping hammer and ultrasonic impact peening) were carried out, providing technical support for the further application of mechanical shock treatment in the field of 2205 duplex stainless steel welding.

## 2. Material and Methods

### 2.1. Material

Commercial 2205 duplex stainless steel with a thickness of 6 mm was selected as the research object, and the ER2209 welding wire with a diameter of 2.5 mm was used for manual argon arc welding. The Ni content of ER2209 welding wire was about 3% higher than that of the base material, which was to promote the transformation of ferrite to austenite in the microstructure of the weld to achieve the purpose of double phase balance of the welded joints. Their chemical composition was determined by the ICP method, as shown in Table 1.

### 2.2. Experimental Methods

Manual argon arc welding of 2205 duplex stainless steel plate was carried out using a two-layer welding process, DC reverse connection, current of 100–130 A, voltage of 9–13 V, and welding speed of 80–110 mm·min^−1^. Before welding, a “V”-shaped groove was opened at the butt joint of the two steel plates. The schematic diagram of the 2205 duplex stainless steel welded joint is shown in Figure 1. The styles were divided into three groups, and sample #1 was not subjected to mechanical shock treatment. Samples #2 and #3 were treated with a pneumatic chipping hammer (PCH) and ultrasonic impact peening (UIP), respectively, when the temperature of the welding layers was less than 100 °C. PCH treatment refers to the use of pneumatic chipping to continuously hammer the welding interface of the sample, resulting in plastic deformation of the surface and residual compressive stress. UIP refers to the use of ultrasonic continuous impact on the surface plastic deformation to produce residual compressive stress.

Pitting corrosion was measured on three groups of samples in different processing states, and the pitting corrosion rate was calculated according to the weight change (weight loss) of the samples before and after the test. U-shaped bending specimens were used for stress corrosion measurement, and the stress corrosion cracking time was obtained. A Gill AC BiSTAT electrochemical workstation was used to test the electrochemical corrosion performance of each group of samples, the medium was 3.5% NaCl aqueous solution (neutral), and the experimental temperature was 25 °C. A two-electrode system was used for testing with a saturated calomel reference electrode and platinum auxiliary electrode. Each group of samples was used as a working electrode for a 20 mV·min^−1^ test.

Three groups of samples were etched with sodium metabisulfite solution of hydrochloric acid (5 mL H_2_O + 3 mL HCl + 2 g Na_2_S_2_O_5_). Under the JSM-IT200 scanning electron microscope (SEM), the metallographic structure of the weld zone and heat-affected zone of the samples was observed. Each group of samples was cut into 0.5 mm thick slices along the welding seam direction, mechanically ground and polished to about 50 μm, and then Φ 3 mm slices were punched out and thinned on the Gatan691 thinner. The microstructure of the samples was observed on the JEM-2010 transmission electron microscope (TEM) at an electron accelerating voltage of 200 kV.

A Stress 3000 X-ray residual stress analyzer was also used to check the residual stress of each group of samples with different treatment methods. Before the test, the equipment was calibrated with standard samples and then the sample was tested according to GB/T7704-2017 (Non-destructive testing—Practice for residual stress measurement by X-ray). The test parameters were the MnKα line, diffraction crystal plane γ-Fe (311) crystal plane, acceleration voltage of 30 kV, acceleration current of 6.6 mA, and exposure time of 10 s. The residual stress was defined as σ_x_ along the weld direction and σ_y_ perpendicular to the weld direction.

## 3. Results and Discussion

### 3.1. Residual Stress

Figure 2 shows the residual stress of 2205 duplex stainless steel welded joints with different treatment methods. As can be seen from Figure 2, untreated sample #1 showed a residual tensile stress of 349 MPa in the X direction and a residual compressive stress of −100 MPa in the Y direction. After PCH and UIP, samples #2 and #3 showed the compressive stress state along the X and Y directions. Residual compressive stress on the surface can effectively slow down the initiation and propagation of cracks and improve the fatigue life of the workpiece [18,19]. Compared with sample #1, the residual compressive stress of sample #2 in the X direction and Y direction reached−281 MPa and −351 MPa, respectively. The residual compressive stress of sample #3 s increased more significantly, reaching −509 MPa in the X direction and −579 MPa in the Y direction. This is because PCH and UIP can fully extend the weld filler metal and reduce the residual tensile stress in the cooling and shrinkage process. Meanwhile, a small amount of plastic deformation on the surface of the weld leads to lattice distortion of the microstructure, which produces beneficial compressive stress on the surface of the weld and eliminates residual tensile stress of the weld. Surface residual tensile stress can greatly shorten the fatigue life of the workpiece. When the workpiece is subjected to external force, the interaction between the external stress and residual stress causes stress redistribution in the section and local plastic deformation. Uneven local deformation leads to the initiation and propagation of cracks, and surface tensile stress will accelerate this process and lead to fatigue failure of the workpiece [20,21]. UIP technology has the best effect of eliminating residual tensile stress on the surface and the fatigue resistance of the welded joint was also the best.

### 3.2. SEM Observations

Figure 3 shows the metallographic structure of the 2205 duplex stainless steel weld zone with different treatment methods. Black is the ferrite phase, and white is the austenite phase. Ferrite mainly comes from the solidification process of the liquid phase, while austenite is mainly produced by the solid phase transformation of ferrite. According to the morphology and precipitation location, the austenite phase can be divided into three types: grain boundary austenite, Widmanstatten austenite, and intragranular austenite. Due to the disordered atomic arrangement and high energy at the grain boundary of ferrite, it is the preferred nucleation site for austenite. Grain boundary austenite does not need too high a degree of undercooling to nucleate, so the formation temperature is high. As the cooling progresses, the content of austenite at the grain boundary gradually increases, and the beneficial nucleation sites available at the boundary decrease. Therefore, the new crystal nucleus will grow rapidly from the boundary into the ferrite grain in the form of side laths, which is called Widmanstatten austenite [22]. If the cooling rate is too fast after welding, the formation of Widmanstatten austenite will be inhibited due to an insufficient time for nucleation and growth. The formation temperature of Widmanstatten austenite is lower, and the content of Cr, Mo, and N is lower than that of grain boundary austenite [23]. Compared with grain boundary austenite and Widmanstatten austenite, the formation of intragranular austenite requires a greater degree of undercooling as a driving force, so the formation temperature is lower. Intragranular austenite usually nucleates in ferrite grains rich in Ni and N elements, but poor in Cr and Mo elements, and the growth of the nucleus is mainly controlled by the subcooling of the composition, so the size is relatively small [22,23]. Under different treatment methods, the metallographic structure of the weld zone of the 2205 duplex stainless steel sample is equivalent. This indicates that the two stress relief methods have no significant effect on the metallographic structure of the 2205 duplex stainless steel weld zone.

Figure 4 shows the metallographic structure of the welded heat-affected zone of 2205 duplex stainless steel under different treatment methods. It can be seen that there is a significant difference between the structure of the welding heat-affected zone and the structure of the base material and weld zone. The structure of the welding heat-affected zone is formed by the rapid heating and cooling process of the base material. Due to the fast cooling rate, the formation of the welding heat-affected zone structure is a non-equilibrium transformation process, and the structure is far from the equilibrium structure [24]. During the welding thermal cycle, the austenite phase in the heat-affected zone begins to transform into ferrite as the temperature increases. The ferrite phase gradually becomes shorter and wider from the original lath shape. As the heating process progresses, the striped austenite phase completely disappears, and the formed ferrite phase is equiaxed. In the subsequent cooling process, the austenite phase first reprecipitates in strips along the ferrite grain boundaries, and grows into the ferrite grains, distributed in blocks or flakes, while the ferrite phase obviously grows isometric crystals. The two stress relief methods had no obvious effect on the metallographic structure of the heat-affected zone of the 2205 duplex stainless steel.

### 3.3. TEM Observations

Figure 5 shows the TEM images of 2205 duplex stainless steel welds with different treatment methods. It can be seen that there are no precipitated phases in the austenite, ferrite, and the boundaries of the austenite–ferrite phase, and the microstructure of each sample is mainly manifested by the difference in dislocation morphology distribution. In the untreated sample #1, the dislocation density in the austenite and ferrite was lower, and the dislocations were mainly distributed at the boundary of the austenite–ferrite phase. After PCH and UIP, there were more dislocations in the austenite and ferrite in samples #2 and #3, but the dislocation density in the ferrite was significantly higher than the austenite. Deformation twins are generated in the austenite phase to coordinate the plastic deformation. Since the stacking fault energy of ferrite is higher than that of austenite, it is easier to form a large number of dislocations in the ferrite during the surface plastic deformation process, causing the proliferation and entanglement of dislocations [25,26]. Among them, the dislocation density of sample #3 treated by UIP was higher.

### 3.4. Pitting Corrosion

Figure 6 shows a histogram of the pitting corrosion rate of 2205 duplex stainless steel welded joints with different treatment methods. It can be seen that after PCH and UIP, the pitting corrosion rate of 2205 duplex stainless steel welded joints was reduced from 1.497 mdd before when untreated to 1.405 and 0.977 mdd, respectively. PCH and UIP technologies improved the distribution of residual stress of the welded joints effectively, and converted the residual tensile stress generated by the phase change during the welding process into the residual compressive stress through the slight plastic deformation generated by the impact. Residual tensile stress can promote the rapid formation and expansion of pitting corrosion holes. As the surface tensile stress is transformed into surface compressive stress, the compactness and uniformity of the surface microstructure are improved, which hinders the progress of pitting corrosion on the surface of the welded joints [27,28]. Among the two mechanical shock treatments, the welded joints treated by UIP had the strongest resistance to pitting corrosion.

### 3.5. Stress Corrosion

Figure 7 shows a histogram of the stress corrosion cracking time of the 2205 duplex stainless steel welded joints with different treatment methods. It can be seen that after PCH and UIP, the stress corrosion cracking time was significantly increased, which showed that the two mechanical shock treatments could significantly improve the stress corrosion resistance of the welded joints. The untreated 2205 duplex stainless steel sample fractured after 4 h under the action of stress corrosion. After PCH and UIP, the stress corrosion cracking time of the samples was increased to 7 h and 13 h, respectively. This was due to the formation of a residual compressive stress layer on the surface of the samples after PCH and UIP. The existence of the residual compressive stress layer hindered the generation and propagation of surface cracks on the samples, and increased the time of stress corrosion cracking [28,29,30]. Among them, UIP technology was the best at improving the stress corrosion performance of 2205 duplex stainless steel welded joints.

### 3.6. Electrochemical Corrosion

Figure 8 shows the polarization curves of the 2205 duplex stainless steel welded joints with different treatments methods, and the measurement results of the electrochemical test is seen in Table 2. The more positive the corrosion potential, the better the corrosion resistance [31,32]. Among all the samples, the corrosion potential of sample #3 treated by UIP was the most positive, the corrosion current density was the lowest, and the pitting potential was the highest, indicating that the corrosion resistance of the 2205 duplex stainless steel welded joints after UIP was the best. Moreover, high pitting potential reduces the pitting rate of chemical corrosion to effectively improve the pitting resistance of welded joints, which is consistent with the trend of the pitting rate in Figure 6. Followed by sample #2, which was treated by PCH, the electrochemical corrosion performance of the welded joints also improved to a certain extent. After PCH and UIP, the residual stress field of the welded joints changed from tensile stress to compression stress. Under the action of residual compressive stress, the distance between atoms in the metal lattice decreased, which led to a decrease in the migration kinetics of point defects through the passivation film, thereby resulting in a low repassivation potential [33]. As the repassivation potential decreased, the corrosion resistance and stability of the passivation film formed on the surface of welded joints became better, which resulted in the improvement of its electrochemical corrosion performance.

### 3.7. Discussion

In conclusion, when the interlayer temperature was lower than 100 °C in the welding process, PCH and UIP treatment in the welding joint area did not change the metallographic structure of the weld seam and the heat-affected zone. However, PCH and UIP treatments changed the surface stress state of the welded joints (from tensile stress to compressive stress) and increased the dislocation density in the phase, resulting in improved stress corrosion resistance, spot corrosion resistance, and strength of the welded joints.

It is common knowledge that the increase in strength is mainly due to hardening of the material as a result of increased dislocation. In the plastic deformation process, slipping is the main deformation mechanism of the ferrite phase due to low stacking fault energy, while the main deformation mechanism of austenite is twinning, due to high stacking fault energy. Both of these can effectively improve the strength of the material. The improvement in stress corrosion performance is mainly due to the surface compressive stress generated by PCH and UIP treatment, which can offset the tensile stress in the welded joint to a certain extent. Surface stress is a very important factor of stress corrosion. If tensile stress is reduced, stress corrosion resistance can be improved. Conversely, the greater the compressive stress, the better the stress corrosion resistance. The surface stress of untreated samples in the X direction and Y direction was tensile stress, while the surface stress of the mechanically treated samples was compressive stress. After PCH treatment, the surface compressive stress in the X direction and Y direction of the sample was −281 MPa and −351 MPa, respectively. After UIP treatment, the surface compressive stress in the X direction and Y direction was −509 MPa and −579 MPa, respectively. Obviously, the compressive stress of samples treated with UIP was greater than that treated with PCH. Therefore, the stress corrosion resistance of specimens treated with UIP and PCH was significantly improved compared with that without treatment, and the specimens treated with UIP were superior to those treated with PCH. PCH and UIP treatments improved the pitting performance by increasing the driving force of surface film damage. For stainless steel, its pitting resistance mainly depends on the destructive power of the film against Cl^−^. The more vulnerable the film is to Cl^−^ damage, the worse its pitting resistance. In the process of welding, PCH and UIP treatment reduced the welding defects and changed the state of surface tensile stress to compressive stress, which increased the driving force of Cl^−^ on film failure. After stress removal, the pitting potential of the specimens was increased to a certain extent, and there was good correspondence between the pitting potential and compressive stress. After UIP treatment, the sample had a larger compressive stress and a higher pitting potential. Therefore, the spot corrosion rate of PCH and UIP treated samples was significantly lower than that of the untreated samples, and the corrosion rate of the UIP treated samples was lower than that of PCH treated samples.

## 4. Conclusions

To conclude, the effects of two kinds of mechanical shock treatments on the residual stress, microstructure, and corrosion resistance of the 2205 duplex stainless steel welded joints were studied and the following conclusions were drawn:Pneumatic chipping hammer and ultrasonic impact peening can effectively improve the stress state of the welded joints from residual tensile stress to residual compressive stress. Compared with the pneumatic chipping hammer, the effect of ultrasonic impact peening was more obvious. The maximum residual compressive stress reached −579 MPa.After pneumatic chipping hammer and ultrasonic impact peening, there were more dislocations in the austenite and ferrite in the samples, and the dislocation density in the ferrite was significantly higher than that of the austenite. Moreover, the dislocation density of welded joints treated by UIP was higher.Pneumatic chipping hammer and ultrasonic impact peening can effectively improve the corrosion performance of the welded joints. Among them, the ultrasonic impact peening technology had the best corrosion resistance of the welded joints.

## Figures and Tables

**Figure 1 materials-15-03230-f001:**
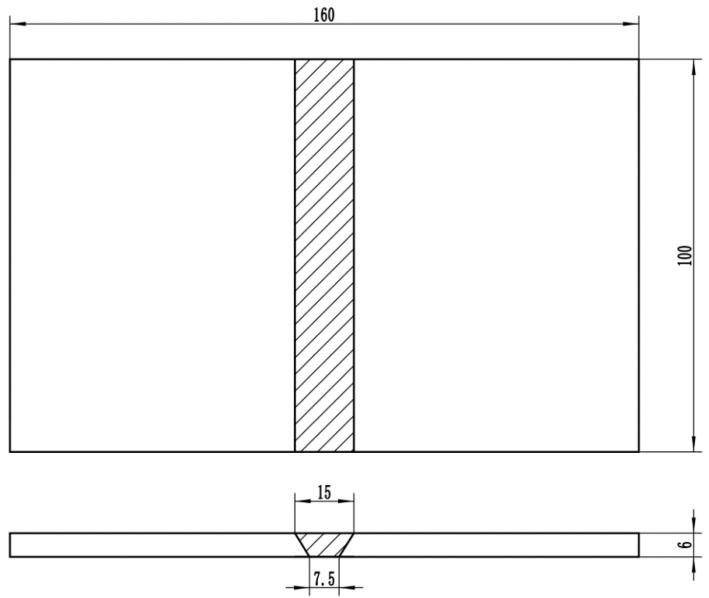
Schematic diagram of 2205 duplex stainless steel welded joint (mm).

**Figure 2 materials-15-03230-f002:**
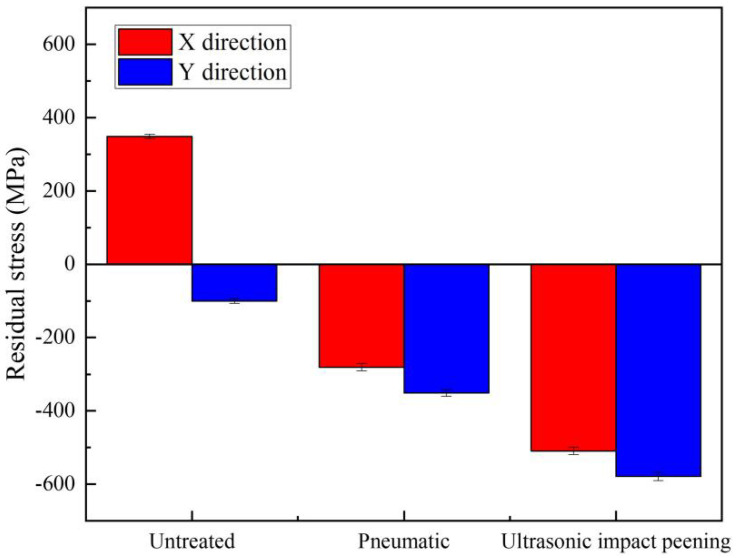
Microstructure of DSS 2205 weld seam area with different treatment methods.

**Figure 3 materials-15-03230-f003:**
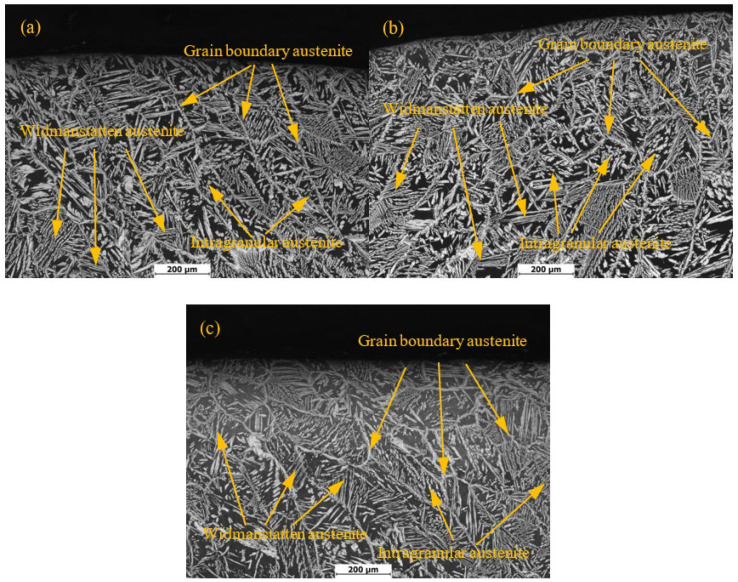
Microstructure of the DSS 2205 weld seam area with different treatment methods: (**a**) Untreated; (**b**) pneumatic chipping hammer; (**c**) ultrasonic impact peening.

**Figure 4 materials-15-03230-f004:**
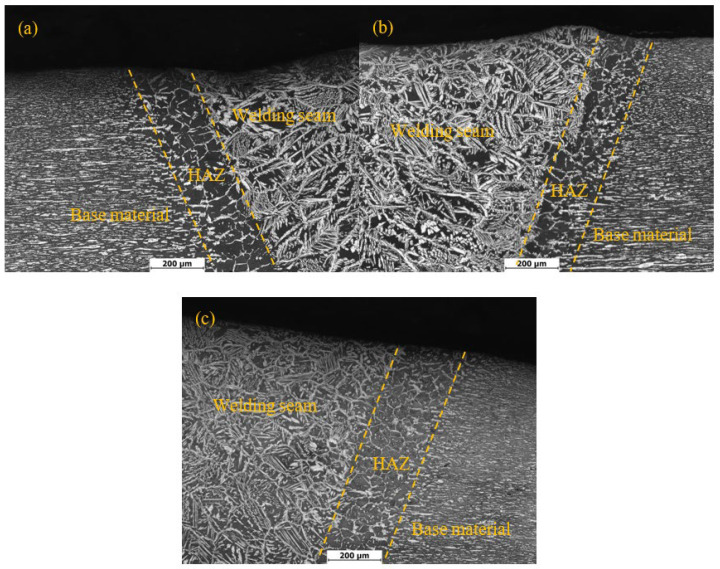
Metallographic structure of DSS 2205 heat affected area with different treatment methods: (**a**) Untreated; (**b**) pneumatic chipping hammer; (**c**) ultrasonic impact peening.

**Figure 5 materials-15-03230-f005:**
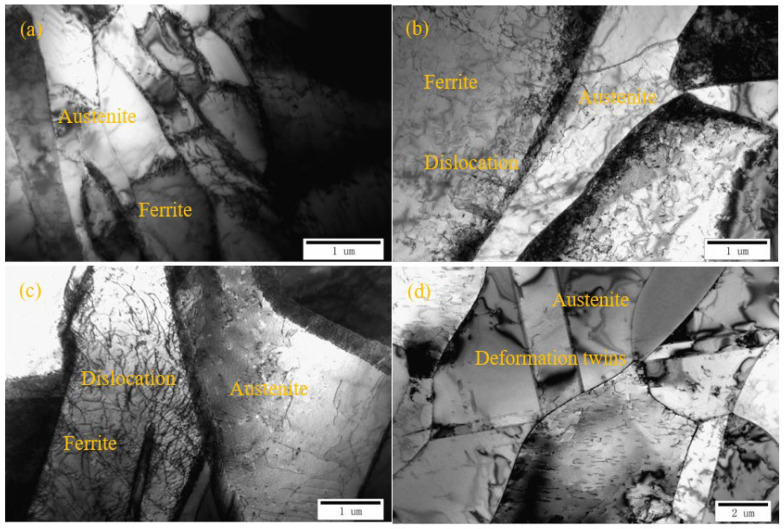
TEM images of DSS 2205 heat affected area with different treatment methods: (**a**) untreated; (**b**) pneumatic chipping hammer; (**c**,**d**) ultrasonic impact peening.

**Figure 6 materials-15-03230-f006:**
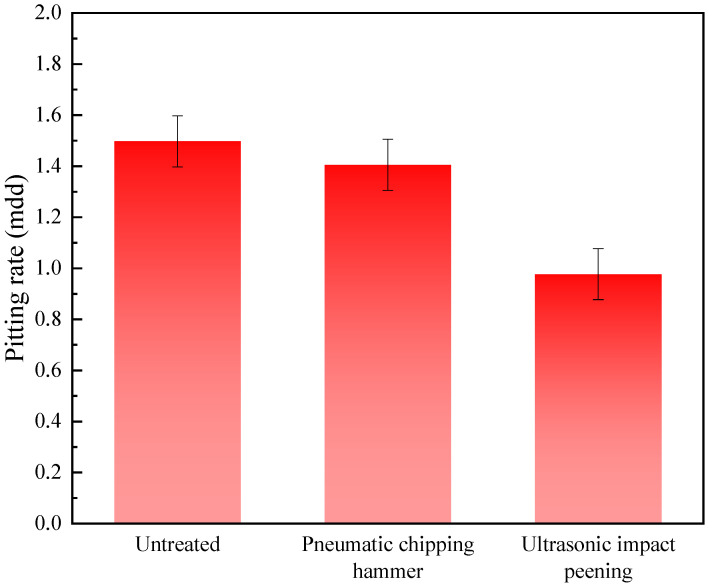
Pitting rate of the welded joints with different treatment methods.

**Figure 7 materials-15-03230-f007:**
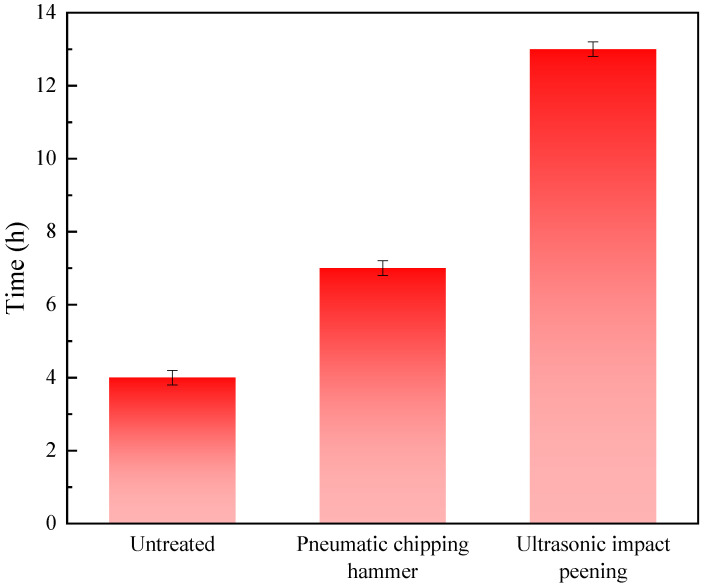
Stress corrosion cracking time of the welded joints with different treatment methods.

**Figure 8 materials-15-03230-f008:**
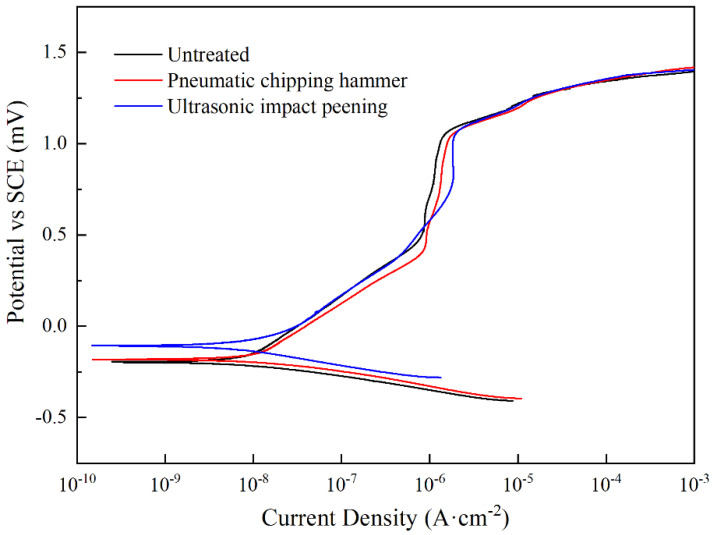
Polarization curve of the welded joints with different treatment methods.

**Table 1 materials-15-03230-t001:** Chemical composition of the 2205 duplex stainless steel and solid wire (wt.%).

Material	C	Si	Mn	Cr	Ni	Mo	N	P	S	Fe
2205	0.025	0.60	1.50	22.5	5.7	3.0	0.15	≤0.03	≤0.03	Balance
ER2209	0.026	0.56	1.74	22.5	8.5	2.9	0.12	≤0.025	≤0.02	Balance

**Table 2 materials-15-03230-t002:** Experimental results of the electrochemistry of welded joint with different treatments.

Treatment	Corrosion Potential (mV)	Corrosion Current Density (mA/cm^2^)	Pitting Potential (mV)	Repassivation Potential (mV)
Untreated	−193	8.8 × 10^−6^	1340	1242
Pneumatic chipping hammer	−179	1.7 × 10^−5^	1347	1159
Ultrasonic impact peening	−110	5.8 × 10^−6^	1355	1180

## Data Availability

Data sharing is not applicable to this article.

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
