# Peer review of "Effect of Mechanical Shock Treatment on Microstructure and Corrosion Properties of Manual Argon Arc Welding Joints of 2205 Duplex Stainless Steel"

_materials, 2022, doi:10.3390/ma15093230_

Round 1

Reviewer 1 Report

You should explain better the essence of your paper, i.e. correlation between residual stress distribution, deformation and dislocation density. It is not clear why would residual stresses change to compressive just due to increase of dislocation density in certain regions, like ferrite? Also explain better what causes what, it is not clear if stress causes dislocation or vice versa, not to mention the role of deformation?

English is not good, especially in the Introduction, I have marked large number of mistakes and phrase which are not understandable. A native English speaker or a professional should carefully correct the whole text.

Author Response

Thank you for your comments.

Reviewer 2 Report

Though the work presented here is interesting, there is no critical analysis as to why PCH and UIP improve the corrosion resistance of the joints. In the current format, the presentation is more like a report rather than a scientific manuscript. My comments are as below:

1.  Discussions section presented by the authors. In the present format, the discussion section does not critically analyze the results. For example, no mechanism or hypothesis is proposed as to why mechanical treatment improves corrosion resistance and other properties.
2. Figure 5: the TEM measurements reported do not have proper diffraction pattern analysis. How can the authors be sure that high dislocation regions are ferrite? A selective area diffraction would prove the author's hypothesis that high dislocation density regions are ferrite.
3. None of the measurements presented in Figure 2, Figure 6, and Figure 7 report the standard deviation of the measurements. This makes one wonder if the authors repeated the experiments to verify the validity of the results? Considering the pitting rate measurements for un-treated and PCH conditions have a closer value. Repeating the experiments and reporting the standard deviation and error associated with the measurements would strengthen the conclusions.
4. What was the rationale behind analyzing the (311) planes for residual stress analysis? Is that the major diffraction plane in the welded samples? Can the authors include raw XRD patterns from all the 3 conditions?
5. What was the reference used for residual stress measurements? Ideally, a stress-free sample with the same composition must be used as a standard to compare the d-spacing changes? It is not clear in the manuscript.

For the above reasons, I recommend a major revision (control missing in some experiments, especially TEM, residual stress analysis, and discussion) for the manuscript.

Author Response

Thank you for your comments.

Reviewer 3 Report

The article has a very interesting topic. Research on the use of treatment other than heat treatment of welded joints allows to set new directions for improving the quality of welds. The material used for testing, i.e. duplex steel, is sensitive to the amount of heat introduced into the weld and the cooling rate. We are not always able to obtain a satisfactory structure - the right proportion of austenite to ferrite.

At work, I found a few things that could be improved:
1. I believe that it would be good to supplement the abstract with the results that can be quantified from the conclusions. For example, an increase in corrosion resistance, etc. This will allow you to gain more interest among potential readers.
2. The acronym table looks strange - please change it
3. There is an erroneous notation of units throughout the work. Please remember that the unit is separated by a space from the numerical value in accordance with the applicable standards. (for example: line 37, 39, 40, 84, 90, 265)
4. line 41 "Daniel et al. [10] studied the laser weld of SAF 2205 duplex stainless steel plate at different welding speeds, and tested the microstructure and mechanical properties of the welded joints. The results show that at different welding speeds, the content of ferrite in the weld zone is all higher than 85%. " - you can also find other works in which other laser welding parameters for ferrite content were checked (e.g. doi:10.2478/adms-2019-0002). When using such a concentrated heat source and a very high cooling rate, it is even impossible to obtain the correct ranges of austenite and ferrite amounts. Please consider including this information for laser welding of duplex steels.
5. Table 1 shows the chemical composition of the additive and base material. Please indicate the source of the chemical composition results. Are these results from manufacturers' approvals or tests performed by the authors? If it was done by the authors, please provide the method, e.g. EDS Analysis.
6. line 87 - "Divide the styles into three groups, 1 # sample was not subjected to mechanical shock treatment during and after welding." - This sentence suggests that some of the samples were machined during welding. On the other hand, in the next sentence it was written that the 2nd and 3rd specimens were mechanically processed after welding and not during the welding.
7. Please describe the procedure for disclosing the microstructure. Report the ratio / concentrations of the etchant. If it was a typical reagent - eg Beraha reagent - the name of the reagent will be sufficient. Time of exposure to etchant, etc.
8. Please provide the manufacturer and country of production of the devices used during the experiment in accordance with the journal's requirements.
9. Fig 3 and 4 - Please provide the exposure parameters for the SEM photos, the detector used for the observation. This is normally included in the microscope photo. This information can be added in the description or in the caption under the drawings.
10. Conclusions could be supplemented with quantifiable effects - not only write about the increase in corrosion resistance, but also write by how many percent the improvement was achieved. Or something similar - this data should also be included in the abstract as I wrote in point 1.
11. The references included several items older than 20 years (even 30 years). Please consider updating these items with possible up-to-date articles. This information can be found in various articles even from last year (Metals and Materials from MDPI, and others)
e.g.: doi.org/10.3390/ma14226791, doi.org/10.3390/met9070762,

Author Response

Thank you for your comments.

Round 2

Reviewer 1 Report

Few sentences are added in an unsuccessful attempt to explain relation between residual stress, microstructure and plasticity, since they are all descriptive. Expressions like "plastic deformation on the surface of the weld leads to lattice distortion of the microstructure..." does not help at all in better understanding of this, the most important aspect of the paper. The paper still does not provide necessary scientific level. Also,  corrections have been made in respect to the review step 1. English has not been improved enough.  

Author Response

First of all, I would like to thank the reviewer for your suggestion again. I can see that you are very rigorous, and I admire you very much!

As mentioned in the last reply, this comment is very good. In this article, the author did not carry out targeted discussion, mainly because the purpose of the work is to provide reference for peer applications, so we did more work at the application level, and seldom considered theoretical knowledge. Considering that residual stress, microstructure and the relationship of plastic are the mature theory of material deformation, so no more redundancy in this aspect, this article just wanted to answer that peening treatment and ultrasonic treatment on the corrosion resistance property has positive effects.

The improving of the language of this article is not enough, which may be because the different language habits, I will carefully revise again, thank you very much again. If it is convenient for the reviewer, you can also specify how to modify in detail. With your help, I believe my language will be greatly improved.

Reviewer 2 Report

The authors have addressed the comments

Author Response

Thanks for your revision!

This manuscript is a resubmission of an earlier submission. The following is a list of the peer review reports and author responses from that submission.

Round 1

Reviewer 1 Report

The paper presents valuable results in the field of welding process of stainless steels.

The effects of the treatments methods, namely: pneumatic chipping hammer and ultrasonic impact peening on the welded samples, are clear presented. The conclusions are sustained by the experimental details and by the results. Ultrasonic impact peening seems to be suitable for stainless steels welding, in order to attained superior corrosion resistance.

In order to be published, I recommend some minor changes, namely:

  • At line 51please specify the load of the HV (HV30…?);
  • In figures 3-5 try to modify the annotations (to be more clear - change color, resolution…)

Best regards!

Author Response

We have revised the manuscript, and I'll give you a reply about your report.

Reviewer 2 Report

small English corrections are required: row 26 and 61 (please check my comments in the enclosed file).

An acronym table or acronym explanation paragraph is required.

The means used to perform UIP and PCH are not described in the paper. The paper should be improved providing some models of the corrosion rate without treatments and after PCH and UIP.

A brief summary about the aim of the paper, its main contributions and strengths:

This paper is related to the experiment done on duplex materials, showing the results and underlining the positive effect of two kind of mechanical treatment for the material behaviour. The authors do not focus very much on the technical demonstration of these positive effects, but they prefer to describe the material behaviour and to show the measures and the microstructure analysis demonstrating that the corrosion rate is lower after the treatments. The paper is well written, even the English is good enough (however you can erase my little comments about the little mistakes that I have found, as this will be better done by your team after my review).

General concept comments

Highlighting areas of weakness, I don't have many concerns about this paper, even looking at it with more criticism (according to the MDPI guidelines for reviewers). What I can say is that the authors have not provided enough technical justifications about the material behaviour after the PCH and UIP. They simply provide the information that the outside layer of the material is harder after the treatment (this could be considered as well known), there is a residual compressive stress in this layer and the dislocation density is changing. Another area of weakness is the modelling of the corrosion behaviour of the material: the authors have not even attempted to provide a draft analysis and the related models of the corrosion rate, the stress corrosion or the electrochemical corrosion (this could be added as point 3.7 in the paragraph 3 of the paper, after the description of the electrochemical corrosion).

Another point that I have underlined is poor description of the mechanical treatments of the samples (this could be added in point 2.2 - R. 88 in the paper). The authors don't describe the tools that they have used to perform PCH and UIP. I think this is important also for the accuracy of the method used and also for the repeatability of the test, which should be relevant for scientific readers.

Then I wish also to underline the importance of an acronym table to explain the main acronym used in the paper. I can mention the units also, because the authors have used mdd units instead of mm, which can be unknown for many readers. I would add this at the beginning of the paper, just after the Keywords.

I don't have other specific comments, referring to line numbers, tables or figures are accurately described and related to the text.

Furthermore:

- The manuscript is clear, relevant for the field and presented in a well-structured manner.

- The cited references are mostly current, 10 of them within the last 5 years, the total number of references is 33. There aren't self-citations.

- The manuscript is enough scientifically sound, but the experimental design is not drawn to test the hypothesis. The better corrosion resistance after the treatments is directly showed by measurements.

- The manuscript’s results can be reproducible after the modification that I have mentioned above (at least the description of the toll system used for the PCH and UIP).

- The figures/tables/images/schemes are appropriate. They properly show the data and they are easy to interpret and understand. The data provided by the authors (not so much data) are interpreted appropriately and consistently throughout the manuscript.

The conclusions are consistent with the arguments presented in the paper.

Author Response

I would give you a reply for your report.

Reviewer 3 Report

Dear authors your paper is interesting and quite well written, however impossible to publish in present form. First of all there are some editorial faults  that have to be corrected, so please read the text carefully one more time. As for merit problem I have few doubt, questions and suggestions.

  1. Fig 1. Present not only the scheme of sample but also its real view,
  2. Line 104, the corrosion medium, why such medium was chosen? This have to be explained. This material have a high corrosion resistance!
  3. How samples were etched? Why there not visible any precipitations (carbides or phases),
  4. Residual stresses , explain why such differences. Have you analyzed surfaces and core or only the core? If only the core why not surface?
  5. Mark description of microstructure in fig. 3.
  6. Present view of microstructure with higher magnification, fig. 3, 4. I do not see the Widmanstatten austenite in those magnification.
  7. In the paper there are lack of critical results discussion. Before conclusions should be made a discussion of all obtained results with explanations why such things appeared. If authors are not sure or literature is non uniform in any field the discussion can contain some hypothesis based on the literature.
  8. Rewrite the conclusion 1.
  9. In general conclusion can be improved because presented results are better than those 3 presented conclusions.

Author Response

I'll give you a reply for your report.

Round 2

Reviewer 3 Report

The paper in this form is possible for publication 

Author Response

Thanks for your reply! I'm very appreciated that you could give me an reply in speciality.